# Physical and Electrical Analysis of Poly-Si Channel Effect on SONOS Flash Memory

**DOI:** 10.3390/mi12111401

**Published:** 2021-11-15

**Authors:** Jun-Kyo Jeong, Jae-Young Sung, Woon-San Ko, Ki-Ryung Nam, Hi-Deok Lee, Ga-Won Lee

**Affiliations:** Department of Electronics Engineering, Chungnam National University, Daejeon 305-764, Korea; jjk1006@cnu.ac.kr (J.-K.J.); sjy5290@o.cnu.ac.kr (J.-Y.S.); kowoon98@cnu.ac.kr (W.-S.K.); nkr0927@cnu.ac.kr (K.-R.N.); hdlee@cnu.ac.kr (H.-D.L.)

**Keywords:** SONOS, flash memory, poly silicon, roughness, data retention, atomic force microscope (AFM), x-ray photoelectron spectroscopy (XPS), deuterium annealing

## Abstract

In this study, polycrystalline silicon (poly-Si) is applied to silicon-oxide-nitride-oxide-silicon (SONOS) flash memory as a channel material and the physical and electrical characteristics are analyzed. The results show that the surface roughness of silicon nitride as charge trapping layer (CTL) is enlarged with the number of interface traps and the data retention properties are deteriorated in the device with underlying poly-Si channel which can be serious problem in gate-last 3D NAND flash memory architecture. To improve the memory performance, high pressure deuterium (D_2_) annealing is suggested as a low-temperature process and the program window and threshold voltage shift in data retention mode is compared before and after the D_2_ annealing. The suggested curing is found to be effective in improving the device reliability.

## 1. Introduction

As the nonvolatile memory market grows rapidly, a lot of research has been reported to improve the device performance and reliability. Especially, 3D silicon-oxide-nitride-oxide-silicon (SONOS) flash memory structure has been suggested to overcome the physical limitation in scaling down the feature size of the existing 2D structure [1,2,3,4,5]. Representative ones are the Stacked Memory Array Transistor (SMArT) by SK Hynix, the Pipe Bit Cost Scalable (P-BiCs) by Kioxia, and Terabit Cell Array Transistor (TCAT) by Samsung. One of the distinct changes in these 3D structures is that a crystalline silicon (c-Si) channel is replaced by a polycrystalline silicon (poly-Si). Poly-Si film is composed of crystalline grains with different crystallographic orientations and grain boundaries with highly defective interfaces [6]. The random mixed structure of grains and grain boundaries is known to cause rough surface compared to a c-Si, which can deteriorate the device performances. In addition, as the area and thickness of the cell decrease due to high integration, the polysilicon channel effect may become more severe. Among the 3D SONOS structures, the devices similar to TCAT structure are based on a gate-last process. That is, a poly-Si channel is first formed and then silicon nitride (Si_3_N_4_) as a charge trapping layer (CTL) is deposited. In this case, the characteristics of the CTL will be affected with the underlayer’s topology because thickness of the CTL is only a few nm. The memory window, date retention as well as program/erase speed of SONOS devices are most affected by trap properties of CTL [7,8,9]. Therefore, when the underlaying poly-Si has large surface roughness as discussed above, the mismatch between materials can be intensified causing larger interface traps and a problem in the device reliability.

In this study, the physical and electrical properties of SONOS device with poly-Si channel are analyzed. Atomic force microscope (AFM) to measure the surface roughness and x-ray photoelectron spectroscopy (XPS) to find out the bonding energy of CTL films with poly-Si underlayer were used. For the electrical analysis, threshold voltage (V_TH_) shift was extracted through the data retention measurements. Moreover, to improve the memory properties, high pressure deuterium (D_2_) annealing is suggested. D_2_ annealing has recently emerged to improve the reliability of MOSFET device by curing shallow traps [10,11]. The experimental results show that by D_2_ annealing the reliability of the SONOS device with poly-Si channel can be improved at low temperature.

## 2. Experiments

A SONOS structured capacitors were fabricated with c-Si and poly-Si as channels. Figure 1 shows the cross-sectional view and process flow of the device. Prime grade p-type c-Si was used as the substrate, and the thickness of tunneling oxide (TO, SiO_2_), CTL (Si_3_N_4_), and blocking oxide (BO, SiO_2_) was 7 nm, 15 nm and 15 nm, respectively. In the case of a poly-Si channel device, 200 nm of thermal SiO_2_ was grown to isolate c-Si and poly-Si, and 50 nm of poly-Si was deposited by low pressure chemical vapor deposition (LPCVD). For TO, c-Si channel device was grown using thermal oxidation and poly-Si channel device was deposited using LPCVD. Then, CTL and BO were deposited using LPCVD. For gate electrode, a 100 nm thick titanium (Ti) film was deposited by RF sputter. Table 1 shows the process conditions of SiO_2_ and Si_3_N_4_ deposited by LPCVD. In this study, high pressure D_2_ annealing is suggested as a passivation method of Si_3_N_4_. High pressure annealing has the advantage of reducing both processing temperature and time. After the gate formation, the annealing was performed at 450 °C, 10 atm, 1 h. The fabricated devices have a gate width by length of 100 μm/100 μm.

## 3. Results and Discussion

### 3.1. Physical Characteristic Analysis

AFM was used to compare the surface roughness of nitride-oxide (NO) stacked film on c-Si and poly-Si. Figure 2a,b shows the AFM images of NO on c-Si, poly-Si. As a reference, the surface roughness of poly-Si on Si substrate is presented in Figure 2c. From Figure 2a,b, it can be seen that Si_3_N_4_ has very rough surface on poly-Si. In Table 2, the extracted roughness values are summarized. From the results, it can be seen that the surface roughness of a film is affected by the underlayer.

From the AFM results, the cross section of the poly-Si device can be drawn as Figure 3, where the roughness of poly-Si causes poor coverage of TO and CTL resulting in the thickness and electric filed fluctuation in each layer. In this case, the electrical characteristics of device are deteriorated by the local increase of electric field in the thin area [12,13]. In addition, the interface roughness influences the interface defect formation by intensifying the lattice constant mismatch between the layers [14,15]. It is well known that the shallow interface traps between TO and CTL affects SONOS flash memory performance [16,17].

#### XPS Analysis

The depth profile analysis of XPS was performed to investigate the bonding structure of the TO and CTL interface according to the channel material change. Figure 4 shows the XPS multi-peak fitting results of Si 2p peak, which are corrected to 285.5 eV of C 1s. The Si 2p spectra can be fitted by 4 peaks using a Gaussian function. Si-Si peak is 99.9 ± 0.15 eV, Si-Si_x_N_y_ peak is 101.3 ± 0.15 eV, Si_3_N_4_ peak is 102.1 ± 0.1 eV, and SiO_2_ peak is 103.4 ± 0.1 eV [18,19]. Si-Si_x_N_y_ bonding represents a combination of Si_3_N_4_ that does not match the composition ratio. Table 3 shows the peak positions and ratios in each device. Ratio is the percentage of the area of each peak to the total area of the peak. The Si-Si and SiO_2_ ratios show similar percentages, but the SixNy and Si_3_N_4_ ratios show opposite results. The reason why the Si-SixNy bonding ratio is higher in poly-Si devices can be explained by the interface degradation as in AFM analysis. Si-Si_x_N_y_ bonding acts as trap in the CTL and affects the reliability of the memory.

Figure 5 shows the basic structure of Si_3_N_4_ and the trap model. In Si_3_N_4_, silicon vacancy (V_Si_) and nitrogen vacancy (V_N_) can be made, and their properties can be changed by atoms entering the vacancy. In general, interface traps have relatively shallower energy traps than bulk traps. As a characteristic of the SONOS structure, since the CTL is adjacent to the oxide layer, oxygen related defects may be formed by oxygen diffusion in the TO or BO. In particular, the bond by the O atom substituted with V_N_ has a small energy level [20]. From the physical analysis, it was confirmed that the roughness deterioration of the underlayer formed more V_Si_ and V_N_ between the TO/BO and CTL, and the increase of the interface trap had a significant effect on the reliability of the memory [20,21,22].

### 3.2. Electrical Characteristic Analysis

#### 3.2.1. Data Retention Measurement

The program (PRG) and data retention behavior of the fabricated devices were measured as shown in Figure 6 and the charge loss in data retention mode was calculated. In data retention mode, C-V were measured after baking at 75~125 °C (25 °C step) for 1 h after programming. Poly-Si channel shows large program window (higher V_TH_ shift) at the same program voltage than c-Si channel. However, larger V_TH_ shift (ΔV_TH_) in the data retention meaning inferior reliability. Figure 7 shows ΔV_TH_ in the data retention mode according to temperature. V_TH_ was extracted as a gate voltage at 80% of the maximum capacitance. ΔV_TH_ is much larger in all temperature conditions and the data retention characteristics are deteriorated in poly-Si devices. Considering shallow traps improves program windows with the traditional tradeoff in data retention properties, the experimental results show that more traps exist in the poly-Si channel device, which is same with the previous physical analysis.

#### 3.2.2. High Pressure D_2_ Annealing Effect

In this study, high pressure D_2_ annealing is suggested to make the device stable as a low temperature process method. Figure 8 shows trap models in Si_3_N_4_. Figure 8a shows 4N-H defect where hydrogen enters into V_Si_. This defect can be ignored because its energy level is not in the bandgap [23]. In Figure 8b, hydrogen enters into nitrogen vacancy (V_N_) and forms a 1Si-H defect and some Si-Si bonds [24,25]. Figure 8c shows O-related traps due to oxygen diffusion into V_N_ [26], which is common near oxide film similar to TO/CTL or CTL/BO interface. In this experiment, to suppress O-related defects near TO/CTL or CTL/BO interface, passivation of V_N_ is focused and high pressure D_2_ annealing is applied as stable curing method at low temperature.

Figure 9 shows the C-V data retention measurement results according to high pressure D_2_ annealing of a poly-Si channel device. Table 4 shows the V_TH_ extracted according to the measurement temperature and high pressure D_2_ annealing. After D_2_ passivation through high pressure annealing, the memory window decreased, but the device reliability was greatly improved. These results are consistent with the previously predicted effect of shallow trap curing of Si_3_N_4_ by D_2_ annealing [27]. This result implies that even though the hydrogen can be dissociated during the post annealing period, some could form stable bondage with Si and N atoms. Furthermore, previous research had conducted D_2_ high pressure annealing even at 900 °C temperature [28].

For the physical analysis on the reduced shallow trap density, it is needed to detect the change in atomic bonding in Si_3_N_4_ by the deuterium bonding. FT-IR analysis can be employed for detecting and determining bond densities of light atoms such as H_2_ or D_2_ [29]. In this experiment, the results are not presented but Thermo-Nicolet 5700 FT-IR spectrometer is analyzed on the c-Si channel device with and without D_2_ treatment where the sample with D_2_ HPA with 600 °C annealing shows slightly increased absorbance in the rage of 2375 cm^−1^. It is difficult to detect the light atoms such as hydron quantitatively, more precise physical method should be studied.

## 4. Conclusions

In this study, the physical and electrical characteristics of the SONOS flash memory device with poly-Si channel were analyzed, and high pressure D_2_ annealing was suggested to improve the device performance. For physical analysis, surface roughness through AFM and trap of TO/CTL interface through XPS were analyzed. From the physical analysis, it was confirmed that underlaying poly-Si deteriorates the surface roughness of CTL and enlarges physical defects. For electrical analysis, ΔV_TH_ was measured in data retention mode and larger in poly-Si devices as confirmed in the physical analysis. However, by high-pressure D_2_ annealing, the deteriorated memory characteristics can be improved. From the data retention measurement before and after D_2_ annealing, it was confirmed that the memory window slightly decreased due to the curing of the interface trap, but the ΔV_TH_ significantly decreased. The results show a problem that appears when poly-Si channel is used in SONOS devices and indicate that high pressure D_2_ annealing is effective method to control the trap sites in interface of CTL.

## Figures and Tables

**Figure 1 micromachines-12-01401-f001:**
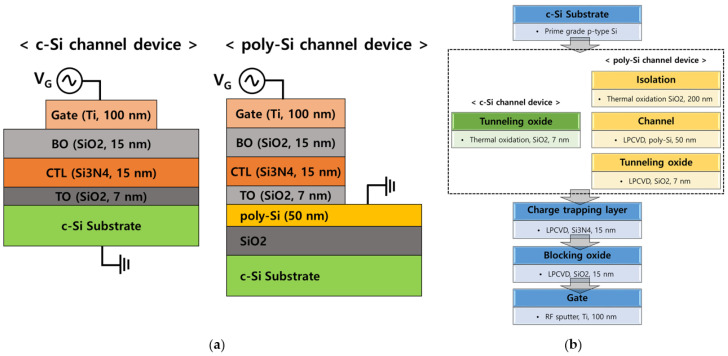
(**a**) The structures of a c-Si and poly-Si SONOS type capacitor devices, and (**b**) the process flow of the device fabrication.

**Figure 2 micromachines-12-01401-f002:**
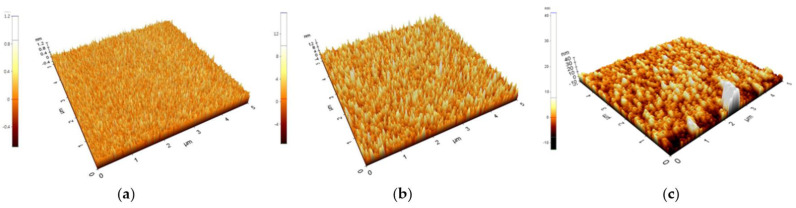
AFM images of fabricated devices in a square region about 5 μm/5 μm. The device structures are (**a**) NO on c-Si, (**b**) NO on poly-Si, and (**c**) only poly-Si deposited on Si substrate, respectively.

**Figure 3 micromachines-12-01401-f003:**
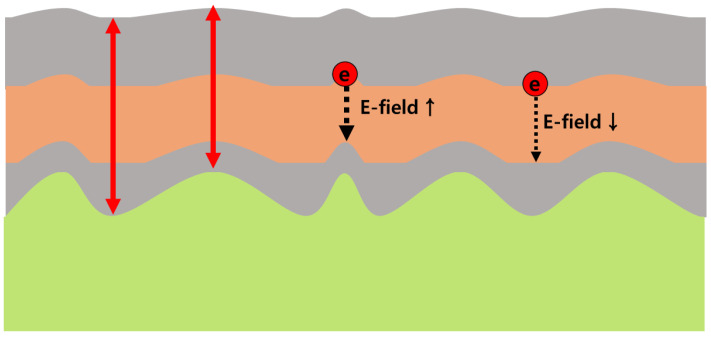
Schematic cross section of a SONOS transistors with poly-Si channel. The roughness of poly-Si causes poor coverage of tunneling oxide and nitride resulting in large electric field fluctuation in each layer.

**Figure 4 micromachines-12-01401-f004:**
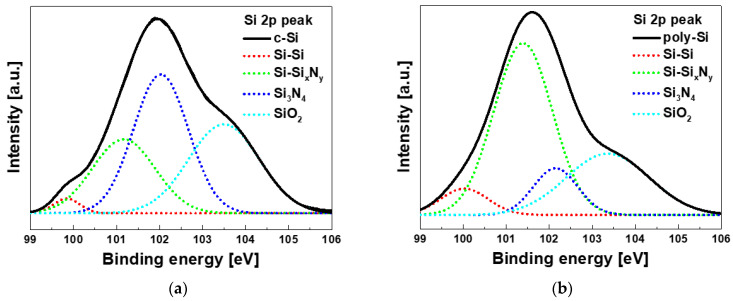
X-ray Photoelectron Spectroscopy (XPS) results of Si 2p multi peak fitting of tunneling oxide/charge trapping layer interface on (**a**) c-Si and (**b**) poly-Si channel device.

**Figure 5 micromachines-12-01401-f005:**
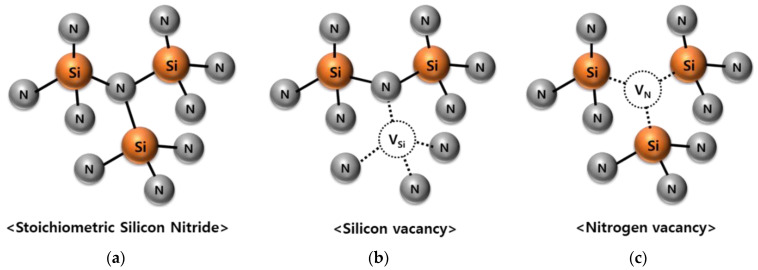
(**a**) Standard structure of the crystal Si_3_N_4_. Main defects (**b**) silicon vacancy and (**c**) nitrogen vacancy.

**Figure 6 micromachines-12-01401-f006:**
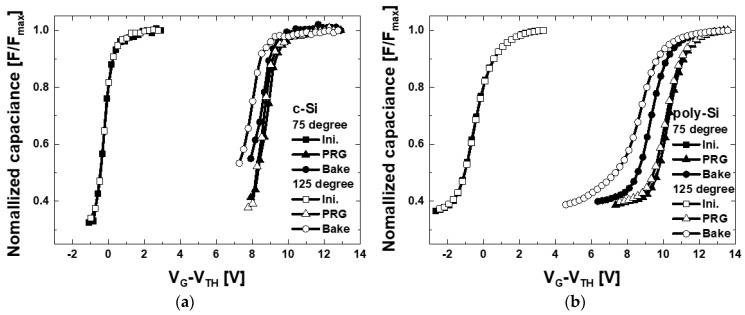
Measurement result of program and data retention characteristics of the fabricated devices. Here, the retention properties were measured after baking at 75~125 °C for 1 h.

**Figure 7 micromachines-12-01401-f007:**
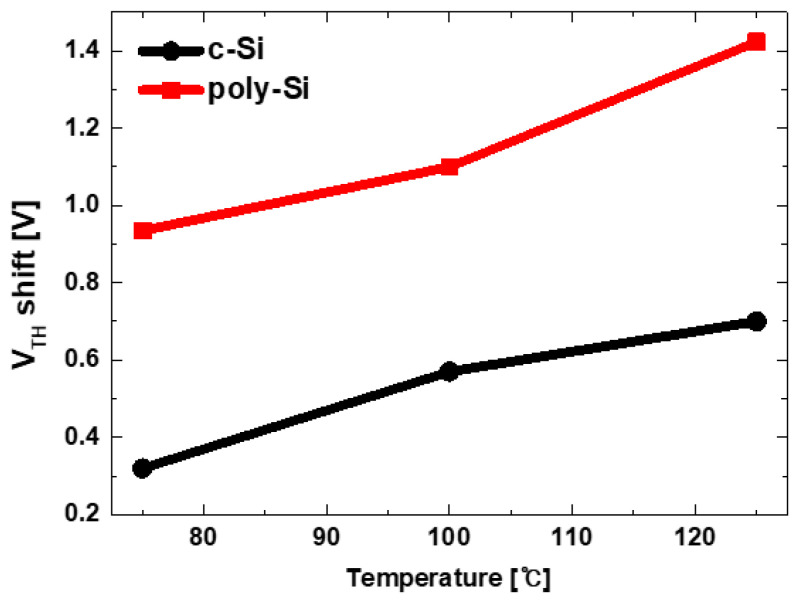
ΔV_TH_ of c-Si and poly-Si channel devices in data retention mode according to the temperature.

**Figure 8 micromachines-12-01401-f008:**
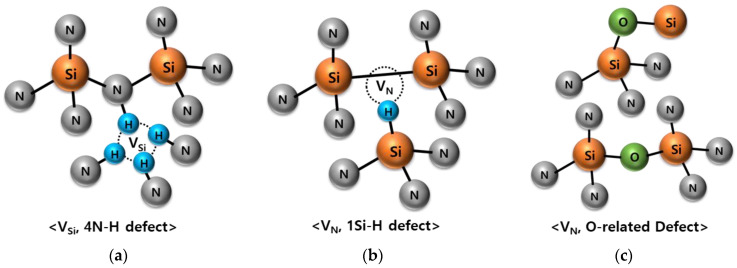
Trap model that can appear in the TO/CTL interface after high pressure D_2_ annealing. (**a**) V_Si_ with substitutional H atom at Si site, (**b**) V_N_ with substitutional H atom at N site or Si-Si bond form, and (**c**) V_N_ with substitutional O atom at N site.

**Figure 9 micromachines-12-01401-f009:**
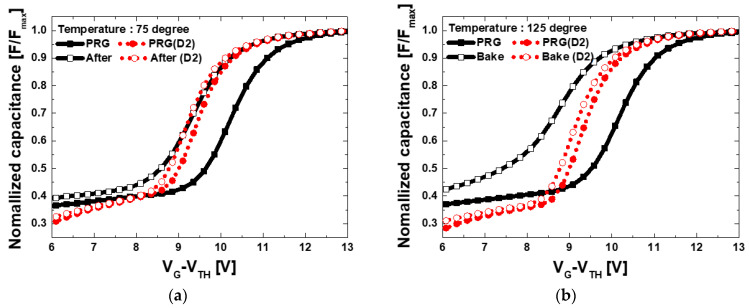
The result of measuring the program and data retention characteristics of the manufactured device according to the high pressure D_2_ annealing of the poly-Si channel device. Data retention measurement temperature (**a**) 75 °C and (**b**) 125 °C.

**Table 1 micromachines-12-01401-t001:** Process conditions of SiO_2_ (BO or TO) and Si_3_N_4_ (CTL) deposited by LPCVD.

LPCVD	Temperature (°C)	Pressure (mTorr)	Composition Ratio
SiO_2_	800	-	Si(OC_2_H_5_)_4_
Si_3_N_4_	720	200	SiH_2_Cl_2_:NH_3_ = 20:120

**Table 2 micromachines-12-01401-t002:** AFM analysis results on surface roughness of NO stacked film on c-Si and poly-Si. In addition, poly-Si roughness is also measured for a reference which is expressed as “Only poly-Si” sample.

Condition	Peak to Valley (nm)	Mean Height (nm)	RMS Roughness (nm)
c-Si channel	1.2	0	0.157
Poly-Si channel	15.715	0.494	2.422
Only poly-Si	12.627	0.158	2.660

**Table 3 micromachines-12-01401-t003:** Extraction results of Si 2p peak parameters according to underlaying channel material.

c-Si	Si-Si	Si_x_N_y_	Si_3_N_4_	SiO_2_	poly-Si	Si-Si	Si_x_N_y_	Si_3_N_4_	SiO_2_
Peak (eV)	99.84	101.17	102.03	103.49	Peak (eV)	100.02	101.39	102.14	103.36
Ratio (%)	2	25	40	33	Ratio (%)	7	54	11	28

**Table 4 micromachines-12-01401-t004:** Extracted V_TH_ based on data retention measurement according to temperature and high pressure D_2_ annealing.

Temperature (°C)	75		125	
Condition	No Treatment	D_2_ Annealing	No Treatment	D_2_ Annealing
Program (V)	10.75	9.86	10.58	9.74
Bake (V)	9.82	9.68	9.16	9.54
ΔV_TH_ (V)	0.93	0.18	1.42	0.20

## Data Availability

Not applicable.

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
