# Peer review of "Physical and Electrical Analysis of Poly-Si Channel Effect on SONOS Flash Memory"

_micromachines, 2021, doi:10.3390/mi12111401_

Round 1
Reviewer 1 Report
This work presents the study result about poly-Si Channel effect with high pressure deuterium(HP-D2) annealing in NAND flash memory. However, what is highlight point(new-fidning) of this paper, compared with other literatures(There are many papers published for HP-D2 in 3D NAND Flash memory)?
Reviewer 2 Report
Submitted manuscript is almost acceptable. The following are comments:
1. Page 5, line133-136: “Considering shallow traps improves pro-gram windows with the traditional tradeoff in data retention properties, the experimental results show that more traps exist in the poly-Si channel device, which is same with the previous physical analysis.”
Does Si3N4 film grown on poly-Si include more amount of Si-H and N-H bonds as a result of termination? Is it possible to evaluate the number of them from XPS or FT-IR spectra? Is larger positive shift caused by cleavage of those bonds and desorption of H atoms from the interface?
2. Figure 9:
During high pressure annealing at 450 degrees, are Si-H bonds at the interface stable?
3. Page 7, line 180-181:”… high pressure D2 annealing is effec-tive method to control the trap sites in bulk or interface of CTL.”
Effectiveness of D2 annealing on bulk properties of the device is not discussed. This part should be deleted or please cite appropriate references to support it.
Reviewer 3 Report
The paper describes the physical and electrical characteristics of the SONOS flash memory device with poly-Si channel. In addition, the high pressure D2 annealing was suggested to improve the device performance.
Overall, the results are technically sound, but the drawn conclusion is not fully supported by the experimental data. The referee recommends this manuscript for being published in the journal micromachines with the major revision. To improve the scientific quality and readability of the manuscript, the referee suggested the following comments:
Major
- The author should define the c-Si. Is it single-crystalline? Is there any difference between c-Si and the substrate? If it is poly-crystalline, is there any difference between c-Si and the relative poly-Si?
- The author uses high-pressure D2 annealing to improve the device performance and predict the reason is H reinforcing defects. However, this prediction is lack of evidence. The author should illustrate the XPS evolution difference after the high-pressure D2 annealing treatment to prove the prediction.
Minor
- In figure 1, the BO, CTL and TO are defined but they aren’t coincident with SONOS. The author should directly add the illustration.
- In line 73, the NO is not defined in the manuscript. I guess the NO same with the NO in SONOS representing nitride-oxide. However, the author still should independently define NO.
- What does the PRG in figure 6 represent? The author should define all the abbreviation clearly in the article.
- In line 174 and 179, the TH in VTH should be subscripted.
Reviewer 4 Report
Accept after minor language-spell checking.
Round 2
Reviewer 1 Report
The highlighted point of the manuscript is not sufficient to be accepted in SCIE Journals. Compared to Ref. (1), (2), the reviewer can not find out which point in this work is the new finding for reliability of poly-Si channel for VNAND? The structure is planar structure and the method for analysis is also too general. In order to get an acceptance, new measurement or analysis technique should be addressed to show the quality of this paper for potential citations.
1. A. Subirats et al,
Impact of discrete trapping in high pressure deuterium annealed and doped poly-Si channel 3D NAND macaroni, 2017 IRPS. (doi: 10.1109/IRPS.2017.7936319)
2. L. Breuil et al,
Improvement of Poly-Si Channel Vertical Charge Trapping NAND Devices Characteristics by High Pressure D2/H2 Annealing, 2016 IMW.
Reviewer 3 Report
The manuscript is now acceptable. By the way, if you want, the vacancy, H or D can be obviously separated by neutron scattering.